# Impact of Low Muscle Mass on Hepatocellular Carcinoma Patients Undergoing Transcatheter Liver-Directed Therapies: Systematic Review & Meta-Analysis

**DOI:** 10.3390/cancers16020319

**Published:** 2024-01-11

**Authors:** Yen-Chun Chen, Meng-Hsuan Kuo, Ching-Sheng Hsu, I-Ting Kao, Chen-Yi Wu, Chih-Wei Tseng, Shih-Chieh Shao

**Affiliations:** 1Division of Gastroenterology, Department of Internal Medicine, Dalin Tzu Chi Hospital, Buddhist Tzu Chi Medical Foundation, Chia-Yi 62247, Taiwan; dm498778@tzuchi.com.tw (Y.-C.C.); hsu.chingsheng@tzuchi.com.tw (C.-S.H.); 2School of Medicine, Tzu Chi University, Hualien 970, Taiwan; 3Department of Pharmacy, Dalin Tzu Chi Hospital, Buddhist Tzu Chi Medical Foundation, Chia-Yi 622401, Taiwan; df441865@tzuchi.com.tw (M.-H.K.); df921180@tzuchi.com.tw (I.-T.K.); df547851@tzuchi.com.tw (C.-Y.W.); 4School of Post-Baccalaureate Chinese Medicine, Tzu Chi University, Hualien 97004, Taiwan; 5Department of Pharmacy, Keelung Chang Gung Memorial Hospital, Keelung 20400, Taiwan; scshao@cgmh.org.tw

**Keywords:** hepatocellular carcinoma, transarterial embolization, transarterial chemoembolization, transarterial radioembolization, sarcopenia, survival, low muscle mass, meta-analysis

## Abstract

**Simple Summary:**

This research addresses the understudied impact of low skeletal muscle mass (LSMM) on intermediate-stage hepatocellular carcinoma (HCC) patients undergoing transcatheter liver-directed intra-arterial therapies. Aiming to determine LSMM’s prevalence and its prognostic significance, the study reveals that 46% of these patients exhibit LSMM, which is consistently associated with decreased overall survival. These findings suggest the need for routine LSMM assessments in clinical settings, potentially influencing treatment strategies and clinical guidelines for HCC management, thus marking a significant contribution to the research community and patient care practices.

**Abstract:**

Background and Aim: Transcatheter liver-directed intra-arterial therapies are mainstream treatment options for intermediate-stage hepatocellular carcinoma (HCC). However, the effect of low skeletal muscle mass (LSMM) on overall survival (OS) in these patients remains uncertain. We aimed to ascertain the prevalence and prognostic effect of LSMM in this population. Method: According to the Preferred Reporting Items for Systematic Reviews and Meta-Analyses guidelines, a comprehensive search was performed in the PubMed and Embase databases until Oct 2023. Random-effects meta-analysis was performed to determine the pooled prevalence of LSMM and calculate the hazard ratio (HR) for OS with a 95% confidence interval (CI) in patients with intermediate-stage HCC undergoing various transarterial therapies, comparing those with and without LSMM. Results: Twelve studies involving 2450 patients were included. The pooled prevalence of LSMM was 46% (95% CI, 38–55%), and the results were consistent across different treatments, regions, and age subgroups. The meta-analysis indicated that LSMM was significantly associated with decreased OS (HR, 1.78; 95% CI, 1.36–2.33; *I*^2^, 75%). Subgroup analyses reassured the main findings across various therapies, including transarterial chemoembolization (TACE) (HR, 1.68; 95% CI, 1.23–2.30; *I*^2^, 81%), transarterial embolization (TAE) (HR, 2.45; 95% CI, 1.42–4.22; *I*^2^, 0%), and transarterial radioembolization (TARE) (HR, 1.94; 95% CI, 1.01–3.73; *I*^2^, 0%). Conclusions: In intermediate-stage HCC, LSMM is common and associated with reduced OS. To achieve an optimal prognosis, clinicians should incorporate routine LSMM measurement into practice, while caring for patients with intermediate-stage HCC, irrespective of TACE, TAE, and TARE.

## 1. Introduction

Hepatocellular carcinoma (HCC), a primary malignant liver tumor predominantly arising in the context of liver cirrhosis, is currently recognized as the fourth leading cause of cancer-related mortality globally [1]. According to the widely recognized Barcelona Clinic Liver Cancer (BCLC) staging framework, HCC identified at the intermediate stage is specifically categorized under BCLC stage B [2,3]. This stage accounts for approximately 30% of all patients with HCC, and is characterized by multifocal tumors, preserved liver function, normal performance status, and an absence of extrahepatic spread or vascular invasion [2,3]. For patients with HCC at BCLC stage B who are not eligible for liver transplantation but still have maintained portal vein flow and a defined tumor burden, consideration of transcatheter liver-directed intra-arterial therapies may be indicated. These treatments cover a range of approaches, namely transarterial chemoembolization (TACE), transarterial embolization (TAE), and transarterial radioembolization (TARE) [1,2,4]. Although TACE is the most frequently used transcatheter liver-directed intra-arterial therapy with survival benefits in patients with non-resectable HCC [4,5], TAE has shown comparable treatment response and survival rates to TACE [6]. Additionally, TARE appears to be a safe alternative treatment [7]. 

Sarcopenia, marked by diminished muscle strength, reduced skeletal muscle mass, and compromised physical performance, is especially common among the elderly and in patients with cancer [8]. The symptoms and side effects of cancer treatments, such as chemotherapy-induced anorexia, tumor burden, and systemic inflammation from the cancer itself, often exacerbate this condition [9]. Additionally, patients with cancer may develop cachexia, marked by increased catabolism, systemic inflammation, and negative energy balance [10]. Despite being distinct conditions, both sarcopenia and cachexia result in muscle loss [11]. Therefore, low skeletal muscle mass (LSMM) is especially common in patients with advanced cancer. Common techniques for evaluating skeletal muscle involve cross-sectional imaging methods, like computed tomography (CT) and magnetic resonance imaging (MRI). A previous meta-analysis suggested that, when assessed by CT imaging, the prevalence of LSMM in patients with HCC undergoing systemic therapies could be as high as 43.2% (95% confidence interval [CI]: 36.3–50.4%) [12]. The exact prevalence in patients with intermediate-stage HCC is still uncertain due to the limited cases reported in previous studies [13,14,15,16,17,18,19,20,21,22]. Since intermediate-stage HCC makes up approximately 30% of all HCC cases, it is essential to understand its prevalence in this population [2,3]. 

Previous studies have reported associations between LSMM and survival outcomes in patients with intermediate-stage HCC undergoing various transcatheter liver-directed intra-arterial therapies, such as TACE, TAE, and TARE [13,14,15,16,17,18,19,20,21,22,23,24]. However, the results of these studies have been inconsistent. Although two studies found no association between baseline LSMM and overall survival (OS) in patients with HCC undergoing TACE [15,17], others have suggested that LSMM is associated with a marked reduction in OS [13,20,21,22,23,24]. The complexity and variability of these findings make it challenging to draw definitive conclusions based on any single study. Therefore, this prompted us to conduct the first systematic review and meta-analysis aimed at summarizing the clinical impact of baseline LSMM on the prognosis of patients with HCC undergoing various transcatheter liver-directed intra-arterial therapies. This comprehensive analysis is intended to provide clearer insights into the prognostic value of LSMM in this population.

## 2. Materials and Methods

### 2.1. Search Strategy

This systematic review and meta-analysis was conducted in strict adherence to the guidelines outlined in the Preferred Reporting Items for Systematic Reviews and Meta-Analyses (PRISMA) statement (refer to Appendix A) [25]. Furthermore, the review followed a protocol that was pre-registered on INPLASY, bearing the registration number INPLASY202380060.

Our research focused on exploring the correlation between LSMM and survival outcomes in patients with HCC undergoing transcatheter liver-directed intra-arterial therapies. We conducted a comprehensive literature search in the PubMed and Embase databases, covering all studies published up until October 2023 [26]. A free-text search strategy was employed using relevant Medical Subject Headings (MeSH) and Emtree terms related to LSMM and liver cancer. In addition to these electronic database searches, we also carried out a manual examination of the reference lists in key original studies and review articles to identify any additional relevant publications. It is noteworthy that our search process did not impose any language restrictions, ensuring a broad and inclusive literature review. Details of our search strategy are reported in Appendix A.

### 2.2. Inclusion and Exclusion Criteria

The inclusion criteria for studies in this systematic review and meta-analysis were meticulously defined based on the PECOS framework, as follows: (1) Patients: individuals diagnosed with HCC undergoing transcatheter liver-directed intra-arterial therapies, including treatments like TACE, TAE, or TARE; (2) Exposures: LSMM; (3) Comparisons: non-LSMM; (4) Outcomes: OS (represented as hazard ratios [HR] and 95% confidence intervals [CIs]); (5) Study design: cohort studies or cross-sectional studies.

Studies were excluded based on criteria including: (1) non-original article publications; (2) research focusing on hepatic tumors different from HCC; (3) investigations not related to LSMM or muscle mass; (4) study populations treated with therapies other than transcatheter liver-directed intra-arterial approaches; (5) absence of statistical information regarding LSMM’s impact on OS including HRs and 95% CIs; and (6) research involving potentially overlapping study populations [12].

### 2.3. Study Selection and Data Extraction

Duplicate studies were automatically excluded using the EndNote X9 software, and then carefully reviewed manually to ensure accuracy. Subsequently, two independent evaluators (K-IT and W-CY) conducted initial screenings of titles and abstracts to pinpoint studies aligning with the inclusion criteria, followed by thorough full-text reviews to ascertain their ultimate suitability. The concordance rate between the two reviewers was 95% for screening titles and abstracts and 89% for full-text article screening. Whenever there was a divergence of opinions on study selection, a third evaluator (H-CS) was engaged to resolve the discrepancies and finalize the decisions.

Two authors, C-YC and K-MC, worked independently to extract key details such as the first author’s name, country of origin, study setting, patient count, sex distribution, participant age, the types of treatments used, the method for estimating muscle mass, the cut-off value for LSMM, the duration of the study, and statistical information regarding the impact of LSMM on OS (including any factors used for adjustment). Any differences in opinions or findings were addressed and resolved through mutual discussions.

### 2.4. Assessment of Methodological Quality 

Research quality was independently assessed using the Newcastle–Ottawa Scale (NOS) by two investigators, K-IT and W-CY [27]. For the NOS, a score of 9 stars indicated a low risk of bias, while scores of 7 or 8 stars denoted a moderate risk of bias [28]. Studies that scored ≤ 6 stars were considered to have a high risk of bias. The NOS demonstrated comparable reliability to other tools used in assessing the risk of bias, and it has been widely employed to evaluate the methodological quality of observational studies in previous systematic reviews and meta-analyses [12].

### 2.5. Statistical Analyses

All statistical analyses were conducted with the aid of Comprehensive Meta-Analysis software, version 4.0 (Biostat, Englewood, NJ, USA) and Review Manager, version 5.3 (Cochrane Collaboration, London, UK, 2020). The prevalence of LSMM in patients with HCC was pooled using a meta-analysis of single proportions. We utilized a random-effects meta-analysis to assess OS in patients with and without LSMM, using adjusted HR values (or unadjusted HR for studies lacking adjusted data) along with 95% CIs. Subgroup analyses were conducted to assess variations in outcomes based on several key parameters. These included treatment modalities, specifically TACE, TAE, and TARE; geographical regions, categorizing studies as either Asian or non-Asian; patient age, with groups divided into those aged 65 years and older (≥65 years) and those younger than 65 years (<65 years); and methods of muscle mass estimation, which included skeletal muscle index (SMI), psoas muscle index (PMI), and fat-free mass index (FFMA). A test for subgroup differences was conducted to compare variations between various subgroups. We considered a *p*-value of <0.1 in this test to indicate statistical difference of effect size within the subgroups [29]. We employed the *I*^2^ statistic to evaluate statistical heterogeneity among the included studies, and either funnel plots or the Egger’s test to assess potential publication bias. To gauge the robustness of our primary analyses, we conducted a sensitivity analysis using a leave-one-out meta-analysis approach. A two-tailed *p*-value < 0.05 was considered statistically significant. 

## 3. Results

### 3.1. Literature Search and Study Selection

We identified 1136 relevant publications, and after removing 210 duplicates, 926 studies were evaluated. Further screening based on titles and abstracts resulted in the assessment of 129 full-text publications for eligibility. Out of the 129 publications, 117 were excluded for not meeting the inclusion criteria: 15 were research focusing on hepatic tumors different from HCC, three were unrelated to LSMM or muscle mass, and 99 involved populations treated with therapies other than transcatheter liver-directed intra-arterial approaches. As a result, 12 studies were ultimately included in the final analysis. The detailed reasons for the exclusion are outlined in Appendix A, and the process and flow diagram for study selection is presented in Figure 1.

### 3.2. Study Description 

This meta-analysis comprised 12 retrospective studies that encompassed 2450 HCC patients who underwent transcatheter liver-directed intra-arterial therapies. All the studies were published between 2018–2023. In this meta-analysis, we utilized twelve prevalence records and conducted eleven OS analyses, as summarized in Table 1. The sample sizes of the individual study cohorts varied, ranging from 58 to 611 participants. Most of the patients were males (n = 1841; 75%). Of the 12 studies included [13,14,15,16,17,18,19,20,21,22,23,24], eight were from Asian countries [13,15,17,19,21,22,23,24], and four were from non-Asian countries [14,16,18,20]. In terms of treatment options, nine studies utilized TACE [13,15,17,19,20,21,22,23,24], two employed TAE [14,18], and one applied TARE [16]. Previous studies have employed different muscle mass measurements. One study reported both SMI and PMI; considering the study’s main result, we included the PMI data into the meta-analysis [22]. Seven records used the SMI [17,18,19,20,21,23,24], two used the PMI [13,15], and two used the FFMA [14,16]. The median observation period was five years (range: 3.3–16.7 years). All the patients had an observation period of >34 months, which was the anticipated median survival time for patients with intermediate HCC following TACE therapy [30]. Regarding the risk of bias, three studies were rated as high [15,17,24] and nine as moderate [13,14,16,18,19,20,21,22,23] (Appendix A).

### 3.3. Prevalence of LSMM in Patients with HCC Treated with Transcatheter Liver-Directed Intra-Arterial Therapies

The prevalence of LSMM was investigated across 12 records, encompassing a total of 2450 individuals [13,14,15,16,17,18,19,20,21,22,23,24]. The range of prevalence was between 30–85%. The reported prevalence rates varied significantly, ranging from 30% to 85%. The overall pooled prevalence of LSMM was determined to be 46%, with a 95% CI of 38–55% and high heterogeneity (*I*^2^ = 94.1%, *p* < 0.001). The Egger’s test for publication bias yielded a *p*-value of 0.38. These findings are graphically illustrated in Figure 2 and Appendix A.

Subgroup analyses revealed a consistently high prevalence of LSMM across different treatment modalities: 43% (95% CI, 40–45%) in TACE, 71% (95% CI, 30–93%) in TAE, and 30% (95% CI, 22–41%) in TARE. In a sub-analysis involving data from Asian individuals, the prevalence was 41% (95% CI, 33–49%). In non-Asian individuals, the prevalence was 58% (95% CI, 35–78%). Additionally, the prevalence was 48% (95% CI, 33–62%) in patients ≥ 65 years of age and 45% (95% CI, 34–56%) in patients < 65 years. The prevalence rates were as follows: 48% (95% CI, 34–62%) for studies that defined LSMM using SMI, 47% (95% CI, 43–51%) for studies using PMI, and 40% (95% CI, 23–60%) for studies using FFMA (Appendix A).

### 3.4. Overall Survival in Patients with HCC Treated with Transcatheter Liver-Directed Intra-Arterial Therapies with and without LSMM

Eleven records [13,14,15,16,17,18,19,20,22,23,24] involving 2388 patients reported results related to OS. The pooled HR for OS was found to be 1.78, with a 95% CI of 1.36 to 2.33 and a *p*-value of less than 0.001, indicating a statistically significant association. The heterogeneity among these studies was relatively high (*I*^2^ = 75%). Additionally, the Egger’s test for publication bias showed a *p*-value of 0.12. These results are visually represented in Figure 3A and Appendix A.

The results of our subgroup analyses are presented in Appendix A. Adjusted pooled analysis showed a similar association between LSMM and OS. Analyzing data from ten records that included multivariable analysis data, we found an HR of 1.97 (95% CI, 1.44–2.69, *p* < 0.001; *I*^2^, 78%), demonstrating a significant relationship between LSMM and reduced OS. LSMM consistently showed an association with reduced OS across various treatment modalities: for TACE, the HR was 1.68 (95% CI, 1.23–2.30; *I*^2^, 83%); for TAE, the HR was 2.45 (95% CI, 1.42–4.22; *I*^2^, 0%); and for TARE, the HR was 1.94 (95% CI, 1.01–3.73; *I*^2^, 0%). Similar trends were observed in different demographic groups, with the HR for Asian populations being 1.69 (95% CI, 1.19–2.40; *I*^2^, 83%) and for non-Asian populations being 1.97 (95% CI, 1.44–2.70; *I*^2^, 0%). Age subgroup analysis showed an HR of 1.62 (95% CI, 1.31–2.02; *I*^2^, 0%) in patients aged ≥ 65 years and an HR of 1.92 (95% CI, 1.13–3.27; *I*^2^, 92%) in patients aged < 65 years. Pooled data from records of muscle mass measured by SMI (HR, 1.83; 95% CI, 1.20–2.80; *I*^2^, 84%), PMI (HR, 1.44; 95% CI, 1.15–1.82; *I*^2^, 0%), and FFMA (HR, 2.23; 95% CI, 1.36–3.65; *I*^2^, 0%) demonstrated a similar association. These findings are elaborated in Appendix A.

### 3.5. Publication Bias and Sensitivity Analysis

The assessment of publication bias for the prevalence studies was conducted using a funnel plot, which demonstrated a symmetrical distribution, as depicted in Appendix A. This symmetry, corroborated by the Egger’s test with a *p*-value of 0.38, suggests an absence of publication bias. Similarly, the funnel plot evaluating the association between LSMM and OS also exhibited visual symmetry (see Appendix A), further supported by the Egger’s test result (*p* = 0.12).

In addition, a sensitivity analysis was performed by sequentially removing one study at a time. This analysis consistently indicated a significant impact of LSMM on OS, reinforcing the robustness of our findings. The pooled results maintained their statistical significance across various configurations, demonstrating stability and reliability even with the exclusion of any individual study. The detailed outcomes of this sensitivity analysis are illustrated in Appendix A.

## 4. Discussion

This systematic review and meta-analysis illustrate that LSMM is common in patients with HCC undergoing transcatheter liver-directed intra-arterial therapy, with an overall prevalence of 46%. The prevalence rate remained consistently high across various subgroups. Our results suggest that LSMM is associated with decreased OS. This association was evident across different treatments (e.g., TACE, TAE, and TARE), regions (Asian and non-Asian areas), age groups (≥65 and <65 years), and muscle mass measurement methods (SMI, PMI, and FFMA). These results highlight the potential clinical importance of LSMM in the management and prognosis of HCC.

In this meta-analysis, patients receiving transcatheter liver-directed intra-arterial therapies for HCC frequently exhibited LSMM, with an overall prevalence of 46% (95% CI, 38–55%). These results are consistent with previous findings of 43.2% in patients with HCC undergoing systemic therapy (95% CI, 36.3–50.4%; n = 2280) [12] and 41.7% across all stages of HCC (95% CI, 36.2–47.2%; n = 9790) [31]. In contrast, this meta-analysis observed a broad range (between 30–85%) and heterogeneity in prevalence. These variations could be attributed to different ethnic backgrounds, sex ratios, ages, methods used to estimate muscle mass, and LSMM cut-off values. For example, the study by Lanza et al. reported the highest prevalence of LSMM at 85% [18]. This study employed the highest SMI cut-off values (males: <55 cm^2^/m^2^; females: <39 cm^2^/m^2^) and had a higher proportion of males (77%) compared to other studies [18]. Regarding subgroup analysis, the increased prevalence of LSMM in patients treated with TAE was also ascribed to Lanza et al.’s study. Excluding it from our sensitivity analysis, the overall prevalence remained high at 41.6%. Disregarding the study by Lanza et al., the lowest prevalence rate among the included studies was still substantially > 30%. Therefore, the prevalence of LSMM in patients with intermediate-stage HCC is significant and warrants further investigation.

Subgroup analyses showed a consistently high prevalence of LSMM based on the different measurement methods (SMI, PMI, and FFMA). However, the choice of the method for estimating muscle mass can significantly influence these results. Zhang et al. identified a significant linear correlation between PMI and SMI (*p* < 0.001); however, the correlation coefficient was only moderately strong (r = 0.57) [22]. In a study by Bigman et al., three types of measurements were used to define sarcopenia: appendicular lean mass adjusted by body mass index, grip strength, and gait speed [32]. Males were found to have a higher prevalence of sarcopenia based on appendicular lean mass/body mass index and grip strength, but a lower prevalence based on gait speed. Additionally, compared to white non-Hispanics, black non-Hispanics had a lower prevalence of sarcopenia, as measured by grip strength, and a higher prevalence, as measured by gait speed [32]. These variations in detection methods can introduce discrepancies in prevalence assessments. Therefore, the application of standardized methods and cut-off values is crucial for refining further prognostic evaluations.

Several meta-analyses have established a link between LSMM and poor prognosis in patients with HCC [12,33]. This current analysis, which focuses on those undergoing transcatheter liver-directed intra-arterial therapies, reiterates these outcomes, with an overall HR of 1.78 (95% CI, 1.36–2.33) and adjusted HR of 1.97 (95% CI, 1.44–2.69). The reasons for the increased mortality in patients with sarcopenic HCC remain unclear. Increased susceptibility to infection and malnutrition are the potential contributing factors. Sarcopenia-related factors, such as metabolic and hormonal irregularities and circulating endotoxins, increase the risks of infection and chances of death by sepsis [34]. Skeletal muscle mass is a more accurate representation of nutritional status than solely relying on serum albumin levels in patients with cirrhosis and HCC [13]. LSMM indicated that malnutrition results in increased mortality, prolonged hospital stays, and higher medical costs [35]. Therefore, it is reasonable for patients with LSMM to have a poorer prognosis than non-LSMM patients.

A significant observation from this study was the consistent association observed between LSMM and a poorer prognosis in patients with HCC undergoing transcatheter liver-directed intra-arterial therapies. This association persisted across various treatment modalities, study regions, patient age groups, and methods used for measuring muscle mass. Notably, the tests for subgroup differences did not reveal any significant disparities among these subgroups. This uniformity in findings suggests that addressing and potentially improving LSMM could play a crucial role in enhancing the outcomes of HCC management and may serve as a reliable factor in informing treatment decisions. However, the current HCC staging system does not consider LSMM as a determinant factor [2,3]. Therefore, future large-scale prospective studies should evaluate the utility of integrating LSMM into a revised staging system and assess its potential effect on improving the prognosis of patients with HCC.

This study has several limitations. First, the included studies used different methods and cut-off values to determine LSMM using CT or MRI. This variation may have led to differing prevalence rates. However, our subgroup analysis suggested that these methodological differences did not influence the overall effect of LSMM on survival. This suggests that LSMM has a detrimental impact, irrespective of the methodology or threshold values employed. In future studies, standardized assessment methods and thresholds are essential. Secondly, the generalizability of these findings is potentially limited, as the majority of the included studies in this systematic review and meta-analysis were conducted in East Asia. This regional concentration raises questions about the applicability of the results to patients from other geographical areas. To address this limitation and reduce potential biases, there is a compelling need for more globally diverse studies. Such studies should focus on patients with intermediate-stage HCC from various regions to better understand the impact of LSMM on prognosis across different populations. Third, it is important to note that several studies included in this meta-analysis were characterized by limited sample sizes. However, to assess the impact of this limitation, we conducted a sensitivity analysis utilizing a one-study removal approach. The results of this analysis demonstrated that our pooled findings remained robust, indicating that the inclusion of these studies with smaller sample sizes had a minimal effect on the overall conclusions drawn from our meta-analysis. Fourth, since most of the included studies were retrospective and conducted at single centers, they were judged to have a moderate–to–high risk of bias. We must interpret our findings with caution before considering their application in clinical practice.

The primary strength of this systematic review lies in its utilization of updated and diverse data sources, encompassing a wide range of studies that focused on patients with intermediate-stage HCC undergoing various transcatheter liver-directed intra-arterial therapies. This inclusive approach significantly augmented the statistical power of our analysis, thereby facilitating a more comprehensive and nuanced assessment of the impact of LSMM on survival outcomes. Furthermore, our meta-analysis addresses a gap in existing research by performing subgroup analyses to evaluate the impact of various transcatheter liver-directed intra-arterial therapies on patient prognosis in HCC. This method enhances the current knowledge base and offers physicians evidence-based insights to develop prognosis strategies tailored to patients’ muscle mass status.

## 5. Conclusions

Low skeletal muscle mass is common in patients with intermediate-stage HCC, and is associated with a reduced OS. To achieve an optimal prognosis, clinicians should incorporate routine measurement of LSMM into practice while caring for patients with intermediate-stage HCC, irrespective of TACE, TAE, and TARE. Further research is required to establish standardized methods for measuring muscle mass and investigate whether treating sarcopenia or LSMM can enhance survival outcomes.

## Figures and Tables

**Figure 1 cancers-16-00319-f001:**
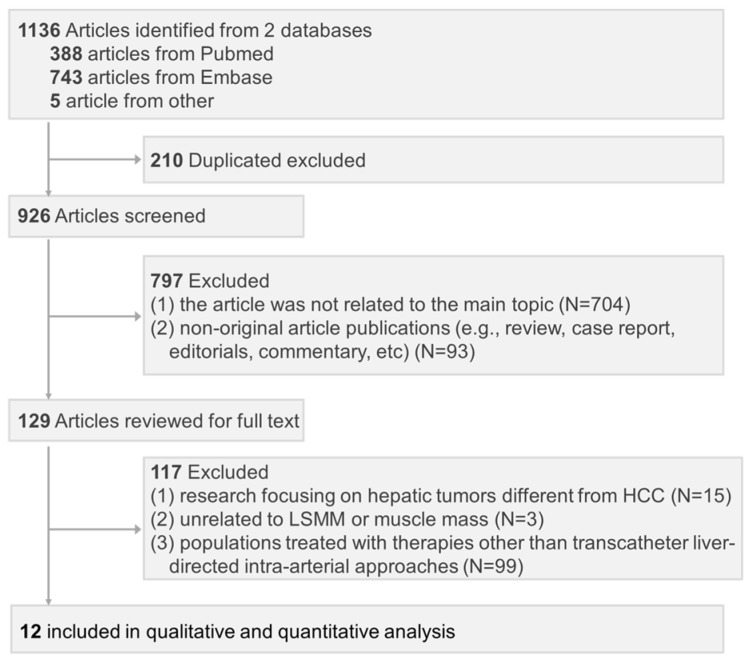
Study selection based on the PRISMA diagram.

**Figure 2 cancers-16-00319-f002:**
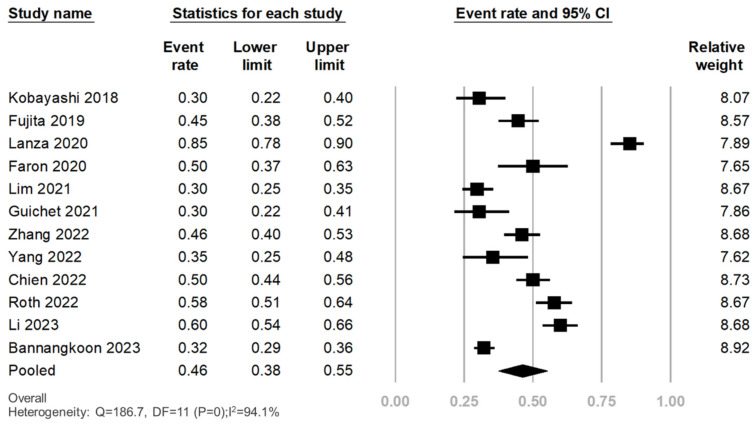
Prevalence of LSMM in patients with HCC undergoing TACE/TAE/TARE [13,14,15,16,17,18,19,20,21,22,23,24]. LSMM, low skeletal muscle mass; HCC, hepatocellular carcinoma; TACE, transarterial chemoembolization; TAE, transarterial embolization; TARE, transarterial radioembolization.

**Figure 3 cancers-16-00319-f003:**
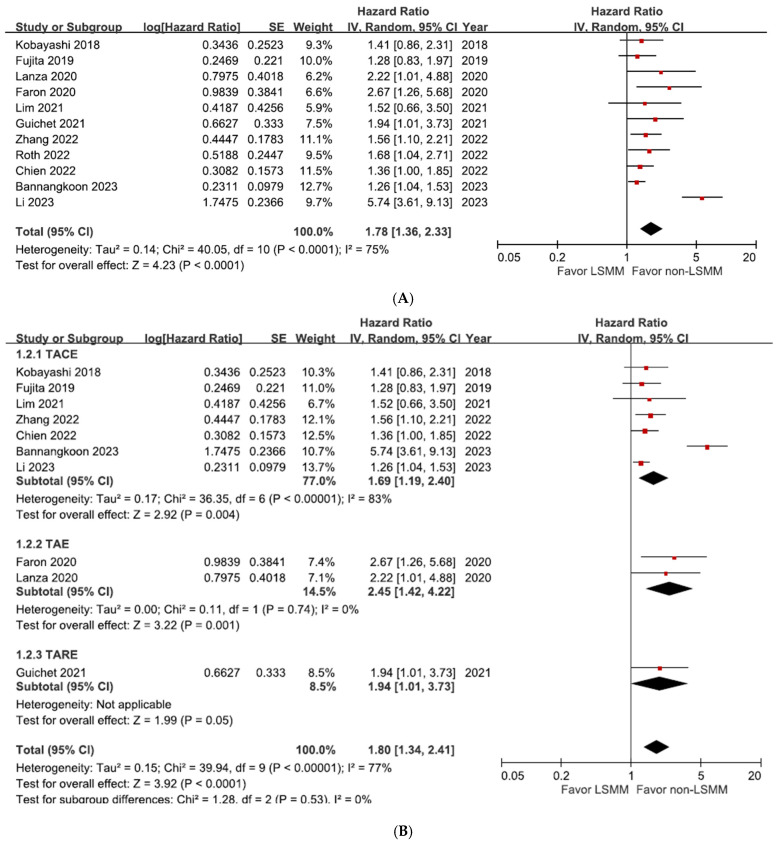
Forest plot illustrating the analysis of the relationship between LSMM and overall survival. (**A**) overall [13,14,15,16,17,18,19,20,22,23,24], (**B**) by treatment [13,14,15,16,17,18,19,22,23,24], (**C**) by muscle measurement methods [13,14,15,16,17,18,19,20,22,23,24].

**Table 1 cancers-16-00319-t001:** Demographic data and characteristics of included studies.

First Author Year	Country	Setting	Patients, n (M/F)	Age, Years	Method Estimated Muscle Mass	Cut-Off Value for Pre-Treatment LSMM	LSMM (%) Yes/No	Study Period (Year)	OS Adjustment Factors	OS HR (95% CI)	RoB
A. TACE											
Kobayashi 2018 [17]	Japan	multi-center	102 (70/32)	68.3 #	SMI	M: 42 cm^2^/m^2^ F: 38 cm^2^/m^2^	30.4% (31/71)	11	univariate	1.41 (0.86–2.29)	high
Fujita 2019 [15]	Japan	single center	179 (130/49)	72 #	PMI	M: <6.0 cm^2^/m^2^ F: <3.4 cm^2^/m^2^	44% (80/99)	8.3	univariate	1.28 (0.83–1.99)	high
Lim 2021 [19]	Korea	single center	266 (187/79)	69.9 *	SMI	M: 49.6 cm^2^/m^2^ F: 43.1 cm^2^/m^2^	29.69% (79/187)	4.1	age, MELD score, size of tumor, albumin, platelet, BCLC, and objective tumor response	1.52 (0.66–1.11)	moderate
Zhang 2022 [22]	China	single center	228 (175/53)	55.6–63 *	SMI, PMI	(PMI) M: 42.28 mm^2^/m^2^ F: 37.42 mm^2^/m^2^	39% (89/139)	3.3	AFP, Child–Pugh class, maximum tumor diameter, metastasis, and BCLC stage	1.96 (1.39–2.78)	moderate
Yang 2022 [21]	China	single center	62 (49/13)	59.4 *	SMI	M: 42 cm^2^/m^2^ F: 38 cm^2^/m^2^	35.4% (22/40)	6.0	AFP	NR	moderate
Chien 2022 [13]	Taiwan	single center	260 (192/68)	64 #	PMI	M: <6.36 cm^2^/m^2^ F: <3.92 cm^2^/m^2^	50% (130/130)	16.7	maximal tumor diameter ≥ 5 cm, multiple tumors, AFP ≥ 200 ng/mL, albumin < 3.5 g/dL, and VP3/4	1.36 (1.00–1.85)	moderate
Roth 2022 [20]	France	multi-center	225 (200/25)	65 #	SMI	M: <50 cm^2^/m^2^ F: <39 cm^2^/m^2^	57.8% (130/95)	5.0	ascites, size of the largest nodule, AFP, Child–Pugh B	1.68 (1.04–2.72)	moderate
Li 2023 [23]	China	single center	235 (173/62)	54 #	SMI	M: <52.4 cm^2^/m^2^ F: <38.5 cm^2^/m^2^	60% (141/94)	5	age, gender, BMI, smoking history, CNLC staging, CTP, MELD, AST, ALT, hemoglobin, A/G, visceral apodosis	5.74 (3.61–9.11)	moderate
Bannangkoon 2023 [24]	Thailand	single center	611 (445/166)	61 #	SMI	M: <36.2 cm^2^/m^2^ F: <29.6 cm^2^/m^2^	32.2% (197/414)	11	age, chronic lung disease, and chronic kidney disease	1.26 (1.04–1.52)	high
B. TAE											
Lanza 2020 [18]	Italy	single center	142 (110/32)	75 #	SMI	M: <55 cm^2^/m^2^ F: <39 cm^2^/m^2^	85% (121/21)	8.3	performance status, previous treatments, and multifocal disease	2.22 (1.01–4.86)	moderate
Faron 2020 [14]	Germany	single center	58 (45/13)	68 *	FFMA	M: 3582 mm^2^ F: 2301 mm^2^	50% (29/29)	4.1	ECOG-PS, and estimated liver tumor burden	2.68 (1.26–5.70)	moderate
C. TARE											
Guichet 2021 [16]	USA	single center	82 (65/17)	65 *	FFMA	M: 31.97 cm^2^ F: 28.95 cm^2^	30.5% (25/57)	5.0	BCLC stage	1.94 (1.01–3.74)	moderate

* median, #; mean, LSMM, low skeletal muscle mass; OS, overall survival; TACE, transarterial chemoembolization; TAE, transarterial embolization; TARE, transarterial radioembolization; M, male; F, female; SMI, Skeletal muscle mass index; PMI, psoas mass index; FFMA, fat-free muscle area; MELD, Model for End-stage Liver Disease; BCLC, Barcelona Clinic Liver Cancer; AFP, alpha-fetoprotein; VP3/4, major venous thrombosis; ECOG-PS, Eastern Cooperative Oncology Group performance; NR, not reported; AST, aspartate transaminase; ALT, alanine transaminase; A/G, albumin-to-globulin ratio; BMI, body mass index, CNLC, China liver cancer staging; CTP, child Turcotte–Pugh; RoB, risk of bias.

## Data Availability

All datasets and analyses are available and can be accessed upon a reasonable request from the corresponding author.

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
