# Peer review of "Impact of Low Muscle Mass on Hepatocellular Carcinoma Patients Undergoing Transcatheter Liver-Directed Therapies: Systematic Review & Meta-Analysis"

_cancers, 2024, doi:10.3390/cancers16020319_

Round 1

Reviewer 1 Report

Comments and Suggestions for Authors

I have no significant objections to the paper. However, it is  well known how sarcopenia in patients with malignant tumors effects survival and outcome. The  average raiting of the paper is due to the fact that this is another publication describing a worse prognosis in patients with HCC and sarcopenia, this time treated  with various methods of arterial embolization

Reviewer 2 Report

Comments and Suggestions for Authors

This article is of great interest since sarcopenia has recently gained lot of attention due to its prognostic role in HCC patients. 

Methodology is well explained, included the process of database search and the assessment of quality. However, in line 105, I would specify if the process of duplicate removal was conducted manually or by automated tools.

Reviewer 3 Report

Comments and Suggestions for Authors

--The findings are not surprising, as the authors acknowledge.  Persons with sarcopenia in the setting of cirrhosis with HCC are likely to fare worse than those without, whether or not they receive TACE, etc.

--Overall, the MS is well-written, and the results are reliable.

--How often was there disagreement between the two primary reviewers, requiring input from the third? Was there eventful unanimity, or was the decision regarding inclusion based upon 2/3 majority?

--Table S1 could be omitted, as it is available elsewhere.
